# DINO: A Conditional Energy-Based GAN for Domain Translation

**Konstantinos Vougioukas, Stavros Petridis & Maja Pantic**
Department of Computing, Imperial College London, UK
{k.vougioukas, stavros.petridis04, m.pantic}@imperial.ac.uk

## Abstract

Domain translation is the process of transforming data from one domain to another while preserving the common semantics. Some of the most popular domain translation systems are based on conditional generative adversarial networks, which use source domain data to drive the generator and as an input to the discriminator. However, this approach does not enforce the preservation of shared semantics since the conditional input can often be ignored by the discriminator. We propose an alternative method for conditioning and present a new framework, where two networks are simultaneously trained, in a supervised manner, to perform domain translation in opposite directions. Our method is not only better at capturing the shared information between two domains but is more generic and can be applied to a broader range of problems. The proposed framework performs well even in challenging cross-modal translations, such as video-driven speech reconstruction, for which other systems struggle to maintain correspondence.

## 1 Introduction

Domain translation methods exploit the information redundancy often found in data from different domains in order to find a mapping between them. Successful applications of domain translation include image style transfer (Zhu et al., 2017a) and speech-enhancement (Pascual et al., 2017). Furthermore, these systems are increasingly being used to translate across modalities in applications such as speech-driven animation (Chung et al., 2017) and caption-based image generation (Reed et al., 2016). Some of the most popular methods for domain translation are based on conditional Generative Adversarial Networks (cGANs) (Mirza & Osindero, 2014). The conditional information in cGANs is used to drive the generation and to enforce the correspondence between condition and sample. Various alternatives have been proposed for how the condition should be included in the discriminator (Miyato & Koyama, 2018; Reed et al., 2016) but the majority of frameworks provide it as an input, hoping that the sample's correlation with the condition will play a role in distinguishing between synthesized and genuine samples. The main drawback of this approach is that it does not encourage the use of the conditional information and therefore its contribution can be diminished or even ignored. This may lead to samples that are not semantically consistent with the condition.

In this paper, we propose the Dual Inverse Network Optimisation (DINO) framework[1] which is based on energy-based GANs (Zhao et al., 2017) and consists of two networks that perform translation in opposite directions as shown in Figure 1. In this framework, one network (Forward network) translates data from the source domain to the target domain while the other (Reverse Network) performs the inverse translation. The Reverse network's goal is to minimize the reconstruction error for genuine data and to maximize it for generated data. The Forward network aims to produce samples that can be accurately reconstructed back to the source domain by the Reverse Network. Therefore, during training the Forward network is trained as a generator and the Reverse as a discriminator. Since discrimination is based on the ability to recover source domain samples, the Forward network is driven to produce samples that are not only realistic but also preserve the shared semantics. We show that this approach is effective across a broad range of supervised translation problems, capturing the correspondence even when domains are from different modalities (i.e., video-audio). In detail, the contributions of this paper are:

---

[1]Source code: https://github.com/DinoMan/DINO

- A domain translation framework, based on a novel conditioning mechanism for energy-based GANs, where the adversarial loss is based on the prediction of the condition.
- An adaptive method for balancing the Forward and Reverse networks, which makes training more robust and improves performance.
- A method for simultaneously training two networks to perform translation in inverse directions, which requires fewer parameters than other domain translation methods.
- The first end-to-end trainable model for video-driven speech reconstruction capable of producing intelligible speech without requiring task-specific losses to enforce correct content.

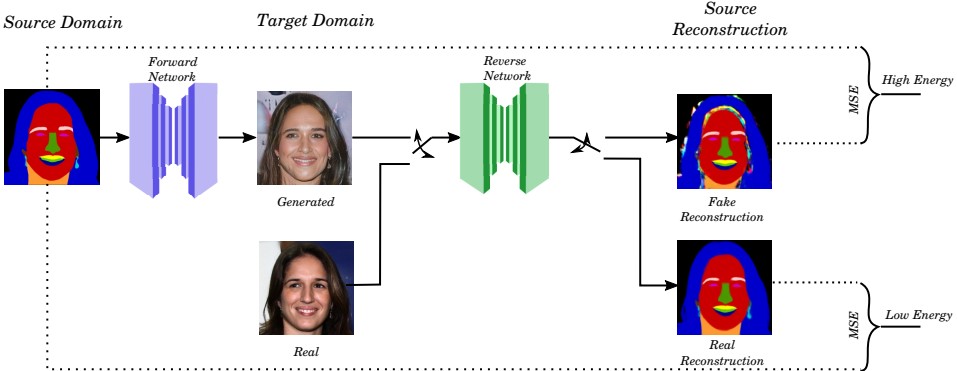

Figure 1: Architecture for the DINO framework. The Forward network performs a translation from the source to the target domain. The Reverse network performs the opposite translation and assigns an energy based on the its ability to recover source domain samples from real and generated samples.

## 2 RELATED WORK

Domain translation covers a wide range of problems including image-to-image translation (Isola et al., 2017), caption-based image synthesis (Qiao et al., 2019), and text-to-speech synthesis (Arik et al., 2017). Unsupervised translation methods attempt to find a relationship between domains using unpaired training data. However, finding correspondence without supervision is an ill-posed problem which is why these methods often impose additional constraints on their networks or objectives. The majority of unsupervised methods are applied to image-to-image translation problems. The CoGAN model (Liu & Tuzel, 2016) imposes a weight-sharing constraint on specific layers of two GANs, which are trained to produce samples from different domains. The motivation is that sharing weights in layers associated with high-level features should help preserve the overall structure of the images. This approach is extended in the UNIT framework (Liu et al., 2017), where the generative networks are Variational Autoencoders (VAEs) with a shared latent space. The weight-sharing used in the CoGAN and UNIT frameworks restricts them to problems where both domains are of the same modality. A more generic method of achieving domain-correspondence is presented in the Cycle-GAN model proposed by Zhu et al. (2017a). The CycleGAN objective includes a cycle-consistency loss to ensure that image translation between two domains is invertible. Recently, Chen et al. (2020) showed that reusing part of the discriminators in CycleGAN as encoders for the generators achieves parameter reduction as well as better results. Although it is possible to apply the cycle consistency loss for cross-modal translation it has not been widely used in such scenarios.

Unlike unsupervised methods, supervised approaches rely on having a one-to-one correspondence between the data from different domains. The Pix2Pix model (Isola et al., 2017) uses cGANs to perform image-to-image translation and has inspired many subsequent works (Zhu et al., 2017a; Wang et al., 2018; Park et al., 2019). Compared to unsupervised methods, supervised approaches have had more success in translating across different modalities. Notable applications include speech-driven facial animation (Vougioukas et al., 2020) and text-to-image synthesis (Reed et al., 2016; Qiao et al., 2019). It is important to note that the adversarial loss in cGANs alone is often not capable of establishing domain correspondence, which is why these approaches also rely on additional reconstruction or perceptual losses (Johnson et al., 2016) in order to accurately capture semantics.

In many scenarios, the relationship between domains is not bijective (e.g. one-to-many mapping) hence it is desirable for translation systems to produce a diverse set of outputs for a given input. Achieving this diversity is a common issue with GAN-based translation systems (Isola et al., 2017; Liu et al., 2017) since they often suffer from mode collapse. The Pix2Pix model (Isola et al., 2017) proposes using dropout in both training and inference stages as a solution to this problem. Another successful approach is to apply the diversity regularisation presented in Yang et al. (2019). Furthermore, many works (Zhu et al., 2017b; Huang et al., 2018; Chang et al., 2018) attempt to solve this issue by enforcing a bijective mapping between the latent space and the target image domain. Finally, adding a reconstruction loss to the objective also discourages mode collapse (Rosca et al., 2017), by requiring that the entire support of the distribution of training images is covered.

## 2.1 CONDITIONAL GANS

The most common method for conditioning GANs is proposed by Mirza & Osindero (2014) and feeds the conditional information as input to both the generator and the discriminator. Using the condition in the discriminator assumes that the correlation of samples with the condition will be considered when distinguishing between real and fake samples. However, feeding the condition to the discriminator does not guarantee that the correspondence will be captured and could even lead to the condition being ignored by the network. This issue is shared across all methods which use the condition as input to the discriminator (Miyato & Koyama, 2018; Reed et al., 2016). Furthermore, it explains why these models perform well when there is structural similarity between domains (e.g. image-to-image translation) but struggle to maintain semantics in cases where domains are significantly different such as cross-modal applications (e.g. video-to-speech).

Another method presented in Park et al. (2019) proposes generator conditioning through spatially-adaptive normalisation layers (SPADE). This approach has been used to produce state of the art results in image generation. It should be noted that this approach requires that source domain data be one-hot encoded semantic segmentation maps and is therefore limited to specific image-translation problems (i.e. segmentation maps to texture image translations). More importantly, conditioning of the discriminator is still done by feeding the condition as an input and hence will have similar drawbacks as other cGAN based methods with regards to semantic preservation.

In some cases it is possible to guide the discriminator to learn specific semantics by performing a self-supervised task. An example of this is the discriminator proposed in Vougioukas et al. (2020) which enforces audio-visual synchrony in facial animation by detecting in and out of sync pairs of video and audio. However, this adversarial loss alone can not fully enforce audio-visual synchronization which is why additional reconstruction losses are required. Finally, it is important to note that finding a self-supervised task capable of enforcing the desired semantics is not always possible.

## 2.2 ENERGY-BASED GANS

Energy-based GANs (Mathieu et al., 2015; Berthelot et al., 2017) use a discriminator $D$ which is an autoencoder. The generator $G$ synthesizes a sample $G(z)$ from a noise sample $z \in \mathcal{Z}$. The discriminator output is fed to a loss function $\mathscr{L}$ in order to form an energy function $\mathcal{L}_D(\cdot) = \mathscr{L}(D(\cdot))$. The objective of the discriminator is to minimize the energy assigned to real data $x \in \mathcal{X}$ and maximize the energy of generated data. The generator has the opposite objective, leading to the following minimax game:

$$\min_D \max_G V(D, G) = \mathcal{L}_D(x) - \mathcal{L}_D(G(z)) \tag{1}$$

The EBGAN model proposed by Mathieu et al. (2015) uses the mean square error (MSE) to measure the reconstruction and a margin loss to limit the penalization for generated samples. The resulting objective thus becomes:

$$\min_D \max_G V(D, G) = \|D(x) - x\| + \max(0, m - \|D(G(z)) - G(z)\|), \tag{2}$$

The margin $m$ corresponds to the maximum energy that should be assigned to a synthesized sample. Performance depends on the magnitude of the margin, with large values causing instability and small values resulting in mode collapse. For this reason, some approaches (Wang et al., 2017; Mathieu et al., 2015) recommend decaying the margin during training. An alternative approach is proposed by Berthelot et al. (2017) which introduces an equilibrium concept to balance the generator

and discriminator and measure training convergence. Energy-based GANs have been successful in generating high quality images although their use for conditional generation is limited.

## 3 METHOD

The encoder-decoder structure used in the discriminator of an energy-based GAN gives it the flexibility to perform various regression tasks. The choice of task determines how energy is distributed and can help the network focus on specific characteristics. We propose a conditional version of EBGAN where the generator (Forward network) and discriminator (Reverse network) perform translations in opposite directions. The Reverse network is trained to minimize the reconstruction error for real samples (low energy) and maximize the error for generated samples (high energy). The Forward network aims to produce samples that will be assigned a low energy by the Reverse network. Generated samples that do not preserve the semantics can not be accurately reconstructed back to the source domain and are thus penalized. Given a condition $x \in \mathcal{X}$ and its corresponding target $y \in \mathcal{Y}$ and networks $F : \mathcal{X} \to \mathcal{Y}$ and $R : \mathcal{Y} \to \mathcal{X}$ the objective of the DINO framework becomes:

$$\min_{R}\max_{F}V(R,F) = \mathscr{L}(R(y),x) - \mathscr{L}(R(F(x)),x), \tag{3}$$

where $\mathscr{L}(\cdot,\cdot)$ is a loss measuring the reconstruction error between two samples. Multiple choices exist for the loss function and their effects are explained in Lecun et al. (2006). We propose using the MSE to measure reconstruction error and a margin loss similar to that used in EBGAN. However, as shown in Mathieu et al. (2015) this method is sensitive to the value of margin parameter $m$, which must be gradually decayed to avoid instability. We propose using an adaptive method inspired by BEGAN (Berthelot et al., 2017) which is based on maintaining a fixed ratio $\gamma \in [0,1)$ between the reconstruction of positive and negative samples.

$$\gamma = \frac{\mathscr{L}(R(y),x)}{\mathscr{L}(R(F(x)),x)} \tag{4}$$

Balancing is achieved using a proportional controller with gain $\lambda$. A typical value for the gain is $\lambda = 0.001$. The output of the controller $k_t \in [0,1]$ determines the amount of emphasis that the Reverse network places on the reconstruction error of generated samples. The balance determines an upper bound for the energy of fake samples, which is a fixed multiple of the energy assigned to real samples. When the generator is producing samples with a low energy they are pushed to this limit faster than when the generator is already producing high-energy samples. Since the ratio of reconstruction errors is kept fixed this limit will decay as the reconstruction error for real samples improves over time. This achieves a similar result to a decaying margin loss without the necessity for a decay schedule. The output of the controller as well as the reconstruction error for real and fake samples during training is shown in Figure 2. We notice that the controller output increases at the start of training in order to push generated samples to a higher energy value and reduces once the limit determined by $\gamma$ is reached. Although this approach is inspired by BEGAN there are some key differences which prevent the BEGAN from working with the predictive conditioning proposed in this paper. These are discussed in detail in Section A.4 of the appendix.

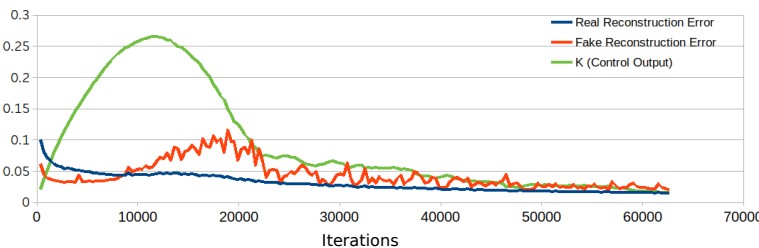

Figure 2: The emphasis placed by the Reverse network (discriminator) on generated samples during training.

In practice we find it advantageous to use the margin loss in combination with adaptive balancing. In this case the margin parameter serves as a hard cutoff for the energy of generated samples and

helps stabilize the system at the beginning of training. As training progresses and reconstruction of real samples improves training relies more on the soft limit enforced by the energy balancing mechanism. In this case we can set $\gamma = 0$ to fall back to a fixed margin approach. The training objective is shown in Equation 5. When dealing with one-to-many scenarios we find that adding a reconstruction loss to the generator's objective can help improve sample diversity.

$$
\begin{cases}
\mathcal{L}_R = \|R(y) - x\| + k_t \cdot \max(0, m - \|R(F(x)) - x\|) \\
\mathcal{L}_F = \|R(F(x)) - x\| \\
k_{t+1} = k_t + \lambda \cdot [\|R(y) - x\| - \gamma \cdot \|R(F(x)) - x\|]
\end{cases}
\tag{5}
$$

### 3.1 BIDIRECTIONAL TRANSLATION

It is evident from Equation 5 that the two networks have different goals and that only the Forward network will produce realistic translations since the Reverse network is trained only using an MSE loss. This prohibits its use for domain translation and limits it to facilitating the training of the Forward network. For the DINO framework, since the Forward and Reverse network have the same structure we can swap the roles of the networks and retrain to obtain realistic translation in the opposite direction. However, it is also possible to train both networks simultaneously by combining the objectives for both roles (i.e. discriminator and generator). This results in the following zero-sum two player game:

$$
\min_R \max_F V(R, F) = \mathcal{L}(R(y), x) - \mathcal{L}(R(F(x)), x) + \mathcal{L}(F(R(y)), y) - \mathcal{L}(F(x), y)
\tag{6}
$$

In this game both players have the same goal which is to minimize the reconstruction error for real samples and to maximize it for fake samples while also ensuring that their samples are assigned a low energy by the other player. Each player therefore behaves both as a generator and as a discriminator. However, in practice we find that is difficult for a network to achieve the objectives for both roles, causing instability during training. The work proposed by Chen et al. (2020), where discriminators and generators share encoders, comes to a similar conclusion and proposes decoupling the training for different parts of the networks. This is not possible in our framework since the discriminator for one task is the generator for the other. To solve this problem we propose branching the decoders of the networks to create two heads which are exclusively used for either discrimination or generation. We find empirically that the best performance in our image-to-image experiments is achieved when branching right after the third layer of the decoder. Additionally, the network encoders are frozen during the generation stage. The bidirectional training paradigm is illustrated in Figure 3.

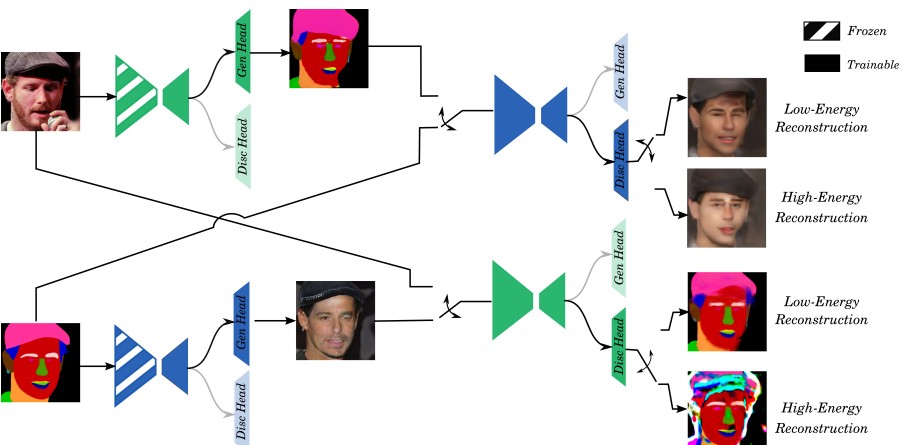

Figure 3: Bidirectional training using the DINO framework. The two players are trained for both the generator and discriminator objectives. The top half of the diagram shows training with the green player as the generator and the bottom of the diagram shows its training as the discriminator. In both cases the blue player assumes the opposite role.

When training network $R$ as a discriminator we use the stream that passes through the discriminative head $R_{disc}$ and when training as a generator we use the stream that uses the generative head

$R_{gen}$. The same applies for player $F$ and which uses streams $F_{disc}$ and $F_{gen}$ for discrimination and generation, respectively. To maintain balance during training we use a different controller for each player which results the objective shown in Equation 7. The first two terms in each players objective represent the player's goal as a discriminator and the last term reflects its goal as a generator.

$$
\begin{cases}
\mathcal{L}_R = \underbrace{\mathscr{L}(R_{disc}(y), x) - k_t \cdot \mathscr{L}(R_{disc}(F_{gen}(x)) - x)}_{\text{discriminator objective}} + \underbrace{\mathscr{L}(F_{disc}(R_{gen}(y)), y)}_{\text{generator objective}} \\
\mathcal{L}_F = \underbrace{\mathscr{L}(F_{disc}(x), y) - \mu_t \cdot \mathscr{L}(F_{disc}(R_{gen}(y)), y)}_{\text{discriminator objective}} + \underbrace{\mathscr{L}(R_{disc}(F_{gen}(x)), x)}_{\text{generator objective}} \\
k_{t+1} = k_t + \lambda_R \cdot [\mathscr{L}(R_{disc}(y), x) - \gamma_D \cdot \mathscr{L}(R_{disc}(F_{gen}(x)), x)] \\
\mu_{t+1} = \mu_t + \lambda_F \cdot [\mathscr{L}(F_{disc}(x), y) - \gamma_G \cdot \mathscr{L}(F_{disc}(R_{gen}(y)), y)]
\end{cases}
\tag{7}
$$

## 3.2 COMPARISON WITH OTHER METHODS

As mentioned in section 2.1 the cGAN conditioning mechanism, used in most supervised translation systems, struggles to preserve the shared semantics in cases where there is no structural similarity between domains. The DINO framework attempts to overcome this limitation by using a different paradigm, where the condition is predicted by the discriminator instead of being fed as an additional input, forcing the generator to maintain the common semantics. Our approach is inspired by several semi-supervised training techniques for GANs (Salimans et al., 2016; Odena et al., 2017; Springenberg, 2015), which have showed that specializing the discriminator by performing a classification task adds structure to its latent space and improves the quality of generated samples. However, these approaches are not designed for conditional generation and use classification only as a proxy task. This differs from our approach where discrimination is driven directly by the prediction of condition.

Another advantage of our system stems from its use of an encoder-decoder structure for the Reverse network. This provides flexibility since the Reverse network can be easily adapted to perform a variety of different translation tasks. In contrast, the multi-stream discriminators used in cross-modal cGANs require fusing representations from different streams. The fusion method as well as the stage at which embeddings are fused is an important design decision that must be carefully chosen depending on the task since it can greatly affect the performance of these models.

The objective of the generator in Equation 5 resembles the cycle-consistency loss used in many unsupervised methods such as CycleGAN (Zhu et al., 2017a) and NICE-GAN (Chen et al., 2020). This also bears resemblance to the back-translation used in bidirectional neural machine translation methods (Artetxe et al., 2018; Lample et al., 2018). However, it is important to note that the cycle-consistency loss used in these approaches is not an adversarial loss since it is optimized with respect to both networks' parameters. The most similar work to ours is MirrorGAN (Qiao et al., 2019), which improves the generation of images through re-description. This model however uses a pre-trained network for re-description in addition to an adversarial loss. Compared to all aforementioned approaches the DINO framework is the only one in which the adversarial loss alone can both achieve sample realism while enforcing correspondence. Finally, since our bidirectional framework uses the generators for discrimination it requires far fewer parameters than these approaches.

## 4 EXPERIMENTS

We evaluate the DINO framework on image-to-image translation since this the most typical application for domain-translation systems. Additionally, we tackle the problem of video-driven speech reconstruction, which involves synthesising intelligible speech from silent video. In all of the experiments focus is placed not only on evaluating the quality of the generated samples but also verifying that the semantics are preserved after translation.

### 4.1 IMAGE-TO-IMAGE TRANSLATION

The majority of modern domain translation methods have been applied to image-to-image translation problems, since it is common for both domains to share high-level structure and therefore easier to capture their correspondence. We evaluate the DINO framework on the CelebAMask-HQ (Lee et al., 2020) and the Cityscapes (Cordts et al., 2016) datasets, using their recommended training-test splits.

When judging the performance of image-to-image translation systems one must consider multiple factors including the perceptual quality, the semantic consistency and the diversity of the generated images. We therefore rely on a combination of full-reference reconstruction metrics and perceptual metrics for image assessment.

Reconstruction metrics such as the peak signal-to-noise ratio (PSNR) and the structural similarity index (SSIM) measure the deviation of generated images from the ground truth. Although these metrics are good at measuring image distortion they are usually poor indicators of realism and they penalize diversity. For this reason, we also measure the perceptual quality of the images by using the Fréchet Inception Distance (FID), which compares the statistics of the embeddings of real and fake images in order to measure the quality and diversity. Furthermore, we use the cumulative probability blur detection (CPBD) metric (Narvekar & Karam, 2009) to assess image sharpness. Finally, we use pre-trained semantic segmentation models to verify that image semantics are accurately captured in the images. For the CelebAMask-HQ dataset we use the segmentation model from Lee et al. (2020) and for the Cityscapes dataset we use a DeepLabv3+ model (Chen et al., 2018). We report the pixel accuracy as well as the average intersection over union (mIOU).

We compare our method to other supervised image-to-image translation models, such as Pix2Pix[2] and BiCycleGAN[3]. Since DINO is a generic translation method comparing it to translation methods that are tailored to a specific type of translation (Yi et al., 2019) is an unfair comparison since these methods make use of additional information or use task-specific losses. Nevertheless, we present the results for SPADE[4] (Park et al., 2019) on the Cityscapes dataset in order to see how well our approach performs compared to state-of-the-art task-specific translation methods. Since the pre-trained SPADE model generates images at a resolution of $512 \times 256$ we resize images to $256 \times 256$ for a fair comparison.

When training the DINO model we resize images to $256 \times 256$ and use a networks with a similar U-Net architecture to the Pix2Pix model to ensure a fair comparison. The architecture of the networks used in these experiments can be found in section A.1.1 of the appendix. Additionally, like Pix2Pix we use an additional L1 loss to train the Forward network (generator), which helps improve image diversity. The balance parameter $\gamma$ is set to 0.8 for image-to-image translation experiments. We train using the Adam optimizer (Kingma & Ba, 2015), with a learning rate of 0.0002, and momentum parameters $\beta_1 = 0.5$, $\beta_2 = 0.999$. The quantitative evaluation on the CelebAMask-HQ and Cityscapes datasets is shown in Tables 1 and 2. Qualitative results are presented in Section A.5.1 of the appendix.

Table 1: Evaluation on CelebAMask-HQ dataset when translating in the direction labels → photos.

| Method | PSNR ↑ | SSIM ↑ | CPBD ↑ | FID ↓ | Pix. Acc. ↑ | mIoU ↑ |
|---|---|---|---|---|---|---|
| Ground Truth | ∞ | 1.00 | 0.54 | - | 93.4 % | 62.1 % |
| Pix2Pix | 11.61 | 0.35 | **0.52** | 59.1 | 93.9 % | 63.1 % |
| BiCycleGAN | 10.47 | 0.31 | 0.50 | 60.4 | 90.5 % | 52.6 % |
| DINO | 12.08 | **0.38** | 0.49 | **51.5** | **96.8 %** | **69.7 %** |
| DINO (Bidirectional) | **12.18** | 0.37 | 0.42 | 55.0 | 96.2 % | 67.3 % |

Table 2: Evaluation on Cityscapes dataset when translating in the direction labels → photos.

| Method | PSNR ↑ | SSIM ↑ | CPBD ↑ | Pix. Acc. ↑ | mIoU ↑ |
|---|---|---|---|---|---|
| Ground Truth | ∞ | 1.00 | 0.72 | 93.9 % | 44.1 % |
| Pix2Pix | 15.96 | 0.37 | 0.68 | 89.4 % | 30.7 % |
| BiCycleGAN | 14.77 | 0.32 | 0.66 | 79.9 % | 21.3 % |
| SPADE | 15.97 | 0.38 | **0.72** | **92.8 %** | **38.3 %** |
| DINO | 16.44 | 0.41 | 0.71 | 91.4% | 35.9% |
| DINO (Bidirectional) | **16.78** | **0.42** | **0.72** | 91.3 % | 35.2 % |

---

[2]`https://github.com/junyanz/pytorch-CycleGAN-and-pix2pix.git`
[3]`https://github.com/junyanz/BiCycleGAN.git`
[4]Pretrained model from: `https://github.com/NVlabs/SPADE`

The results on Tables 1 and 2 show that our method outperforms the Pix2Pix and BicycleGAN models both in terms of perceptual quality as well as reconstruction error. More importantly, our approach is better at preserving the image semantics as indicated by the higher pixel accuracy and mIoU. We notice that for the CelebAMask-HQ dataset the segmentation accuracy is better for generated images than for real images. This phenomenon is due to some inconsistent labelling and is explained in Section A.2 of the appendix. We also note that the bidirectional DINO framework can simultaneously train two networks to perform translation in both directions without a sacrificing quality and with fewer parameters. Finally, an ablation study for our model is performed in A.3 of the appendix.

When comparing our results to those achieved by the SPADE network on the Cityscapes dataset we notice that our model performs similarly, achieving slightly better performance on reconstruction metrics (PSNR, SSIM) and slightly worse performance for preserving the image semantics. This is expected since the SPADE model has been specifically designed for translation from segmentation maps to images. Furthermore, the networks used in these experiments, for the DINO framework, are far simpler (37 million parameters in the generator compared to 97 million). More importantly, unlike SPADE our network can be applied to any task and perform the translation in both directions.

## 4.2 VIDEO-DRIVEN SPEECH RECONSTRUCTION

Many problems require finding a mapping between signals from different modalities (e.g. speech-driven facial animation, caption-based image generation). This is far more challenging than image-to-image translation since signals from different modalities do not have structural similarities, making it difficult to capture their correspondence. We evaluate the performance of our method on video-driven speech reconstruction, which involves synthesising speech from a silent video. This is a notoriously difficult problem due to ambiguity which is attributed to the existence of homophenous words. Another reason for choosing this problem is that common reconstruction losses (e.g. L1, MSE), which are typically used in image-to-image translation to enforce low-frequency correctness (Isola et al., 2017) are not helpful for the generation of raw waveforms. This means that methods must rely only on the conditional adversarial loss to enforce semantic consistency.

We show that the DINO framework can synthesize intelligible speech from silent video using only the adversarial loss described in Equation 5. Adjusting the framework for this task requires using encoders and decoders that can handle audio and video as shown in Figure 4. The Forward network transforms a sequence of video frames centered around the mouth to its corresponding waveform. The Reverse network is fed a waveform and the initial video frame to produce a video sequence of the speaker. The initial frame is provided to enforce the speaker identity and ensures that the reconstruction error will be based on the facial animation and not on any differences in appearance. This forces the network to focus on capturing the content of the speech and not the speaker's identity.

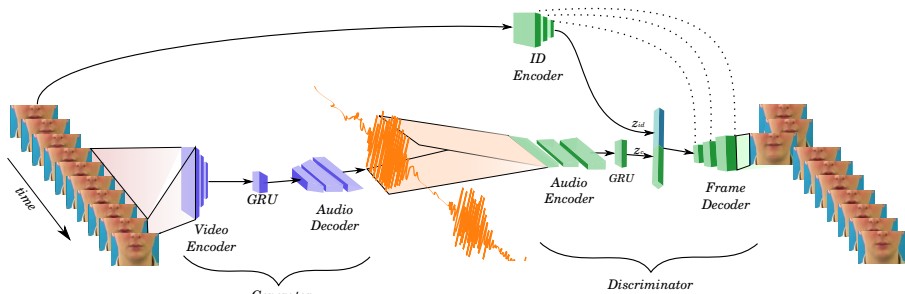

Figure 4: DINO framework architecture used for video-driven speech reconstruction. The Forward network (generator) takes a video as input and outputs a waveform. The Reverse network (discriminator) takes as input a waveform and outputs a video. Components that make up the Forward network are shown in blue and components belonging to the Reverse network are shown in green. More details are shown in Section A.1.2 of the appendix.

Experiments are performed on the GRID dataset (Cooke et al., 2006), which contains short phrases spoken by 33 speakers. There are 1000 phrases per speaker, each containing 6 words from a vocabulary of 51 words. The data is split according to Vougioukas et al. (2019) so that the test set contains unseen speakers and phrases. As baselines for comparison we use a conditional version of Wave-GAN (Donahue et al., 2019) and a CycleGAN framework adapted for video-to-audio translation. Additionally, we compare with the model proposed by Vougioukas et al. (2019), which is designed for video-driven speech reconstruction and uses a perceptual loss to accurately capture the spoken content. An Adam optimiser is used with a learning rate of 0.0001 for the video-to-audio network and a learning rate of 0.001 for the audio-to-video network. The balancing parameter $\gamma$ is set to 0.5.

We evaluate the quality of the synthesized audio based on intelligibility and spoken word accuracy. We measure speech quality using the mean Mel Cepstral Distance (MCD) (Kubichek, 1993), which measures the distance between two signals in the mel-frequency cepstrum and is often used to assess synthesized speech. Furthermore, we use the Short-Time Objective Intelligibility (STOI) (Taal et al., 2011) and Perceptual Evaluation of Speech Quality (PESQ) (Rix et al., 2001) metrics, which measure the intelligibility of the synthesized audio. Finally, in order to verify the semantic consistency of the spoken message we use a pretrained automatic speech recognition (ASR) model and measure the Word Error Rate (WER). The results for the speech-reconstruction task are shown in Table 3.

Table 3: Evaluation of video-driven speech reconstruction on the GRID dataset (unseen subjects).

| Method | PESQ $\uparrow$ | STOI $\uparrow$ | MCD $\downarrow$ | WER $\downarrow$ |
|---|---|---|---|---|
| Ground Truth | 4.5 | 1.00 | 0.0 | 5.4% |
| Conditional WaveGAN | 0.96 | 0.37 | 43.7 | 81.7% |
| CycleGAN | 1.01 | 0.29 | 28.3 | 88.6% |
| Perceptual GAN (Vougioukas et al., 2019) | **1.24** | 0.45 | 24.3 | 40.5 % |
| DINO | 1.21 | **0.51** | **23.0** | **32.6** % |

The results of Table 3 show that our method is capable of producing intelligible speech and achieving similar performance to the model proposed by Vougioukas et al. (2019). Furthermore, the large WER for both baselines highlights the limitations of cGANs and CycleGANs for cross-modal translation. Although our approach is better at capturing the content and audio-visual correspondence, we notice that samples all share the same robotic voice compared to the other methods. This is expected since discrimination using our approach focuses mostly on audio-visual correspondence and not capturing the speaker identity. Examples of synthesized waveforms and their spectrograms are shown in Section A.6 of the appendix and samples are provided in the supplementary material.

**Ethical considerations:** We have tested the DINO model on this task as an academic investigation to test its ability to capture common semantics even across modalities. Video-driven speech reconstruction has many practical applications especially in digital communications. It enables videoconferencing in noisy or silent environments and can improve hearing-assistive devices. However, this technology can potentially be used in surveillance systems which raises privacy concerns. Therefore, although we believe that this topic is worth exploring, future researchers should be careful when developing features that will enable this technology to be used for surveillance purposes.

## 5 CONCLUSIONS

In this paper we have presented a domain translation framework, based on predictive conditioning. Unlike other conditional approaches, predicting the condition forces the discriminator to learn the the relationship between domains and ensures that the generated samples preserve cross-domain semantics. The results on image-to-image translation verify that our approach is capable of producing sharp and realistic images while strongly enforcing semantic correspondence between domains. Furthermore, results on video-driven speech reconstruction show that our method is applicable to a wide range of problems and that correspondence can be maintained even when translating across different modalities. Finally, we present a method for bidirectional translation and show that it achieves the same performance while reducing the number of training parameters compared to other models.

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

## A APPENDIX

### A.1 NETWORK ARCHITECTURE

#### A.1.1 IMAGE-TO-IMAGE TRANSLATION

This section describes the network architecture used for the image-to-image translation experiments in Section 4.1. The two networks used in the DINO framework are identical and both use a U-Net encoder-decoder architecture similar to that used in Pix2Pix (Isola et al., 2017). The encoder is a 7-layer Convolutional Neural Network (CNN) made of strided 2D convolutions. The decoder is a 12-layer CNN made of 2D convolutions and up-sampling layers. We use Instance Normalization (Ulyanov et al., 2016), which has been shown to work well in style transfer applications. The network is shown in detail in Figure 5.

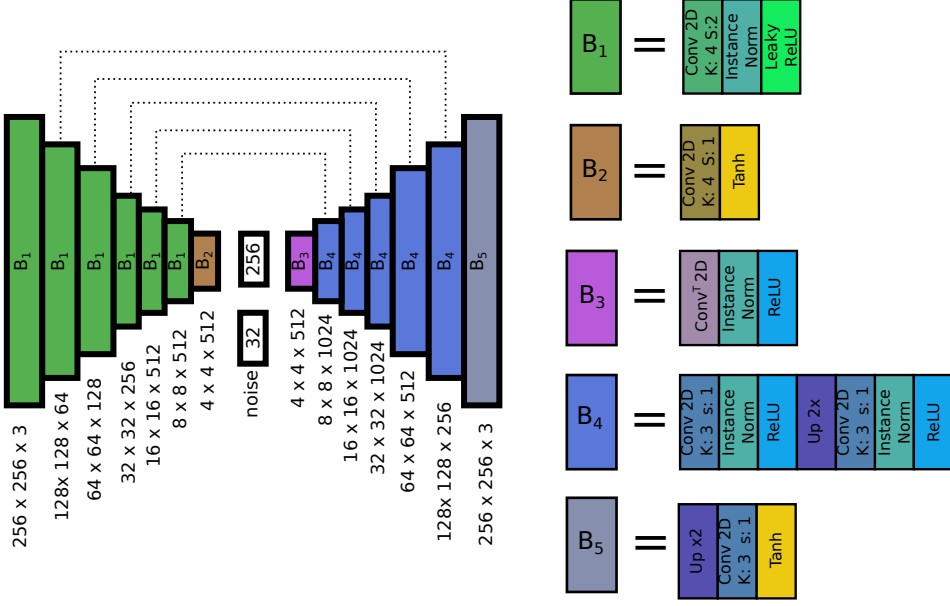

Figure 5: Architecture used in the DINO framework for image-to-image translation experiments.

### A.1.2 VIDEO-DRIVEN SPEECH RECONSTRUCTION

This section describes the architecture of the networks used for video-driven speech reconstruction in the experiments of Section 4.2. In this scenario the Forward network synthesizes speech and the Reverse network performs speech-driven facial animation. The Forward network is made up of a *Video Encoder*, a single-layer GRU and an *Audio Decoder*. The video sequence is fed to the *Video Encoder*, which uses spatio-temporal convolutions to produce an embedding per video frame. The embeddings are fed to a single-layer GRU to create a coherent sequence of representations which is then passed to an *Audio Decoder* network which will produce 640 audio samples per embedding. Concatenating these chunks of samples without overlap forms a waveform. Both the *Video Encoder* and *Audio Decoder* are fully convolutional networks, with the *Audio Decoder* using an additional self-attention layer (Zhang et al., 2019) before the last layer as shown in Figure 6.

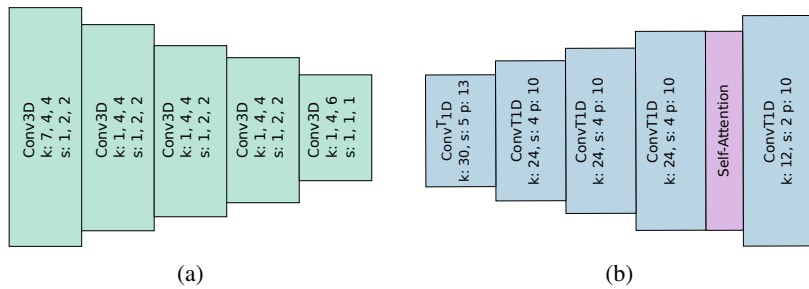

Figure 6: The architecture of the *Video Encoder* (a) and *Audio Encoder* (b) used in the Forward network for video-to-audio translation.

The Reverse network is made up of two encoders responsible for capturing the speaker identity and content. The content stream uses a sliding window approach to create a sequence of embeddings for the audio using an *Audio Encoder* and a 2-layer GRU. The identity stream consists of an *Identity Encoder* which captures the identity of the person and enforces it on the generated video. The two embeddings are concatenated and fed to a *Frame Decoder* which produces a video sequence. Skip connections between the *Identity Encoder* and *Frame Decoder* ensure that the face is accurately reconstructed. A detailed illustration of the Reverse network is shown in Figure 7.

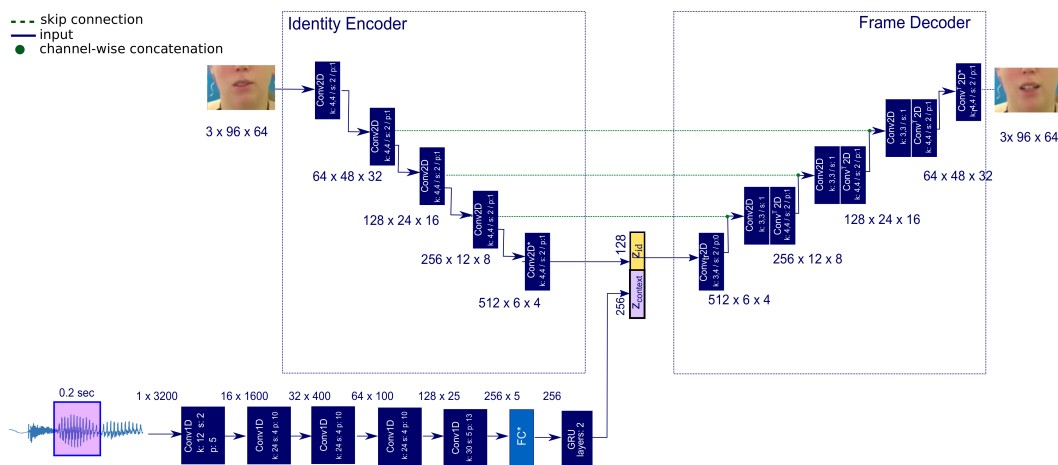

Figure 7: The model for the Reverse network of the DINO framework when performing video-driven speech reconstruction.

### A.2 CELEBA SEGMENTATION

In Table 1 we notice that the segmentation evaluation on generated images surpasses that of real images. The reason for this are some inconsistencies in the labelled images. Examples in Figure 8

show that in these cases some objects are labeled despite being occluded in the real image. However, these objects will appear in the generated images since the labelled images are used to drive their generation. These small inconsistencies in the data annotations explain why segmentation is slightly better for synthesized samples.

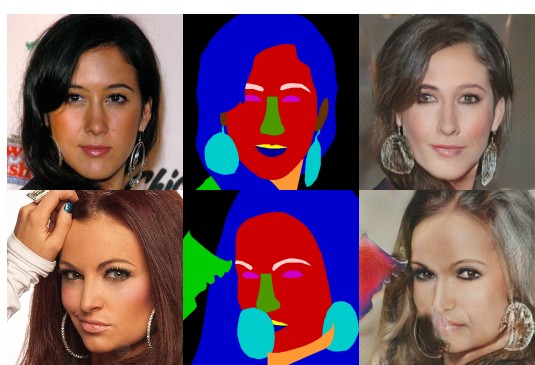

Figure 8: Examples where the annotations in the segmentation map are not consistent with the real image. However, the generated images will remain true to the segmentation map since they are constructed from it.

## A.3   ABLATION STUDY

In order to measure the effect of the reconstruction loss and adaptive balancing used in the DINO framework we perform an ablation study on the CelebAMask-HQ dataset. The results of the study are shown in Table 4. As expected the addition of the L1 loss results in a higher PSNR and SSIM since these metrics depend on the reconstruction error, which is directly optimised by this loss. More importantly, we note that the addition of the L1 loss improves the FID score since it prevents mode collapse. This is evident when observing the examples shown in Figure 9, which shows that mode-dropping that occurs in both DINO and Pix2Pix when this loss is omitted. Finally, we notice that the adaptive balancing used in DINO allows for more stable training and improves performance, which is reflected across all metrics.

Table 4: Ablation study for DINO performed on the CelebAMask-HQ dataset

| Model | PSNR ↑ | SSIM ↑ | CPBD ↑ | FID ↓ | Pix. Acc. ↑ | mIoU ↑ |
|---|---|---|---|---|---|---|
| DINO w L1 ($\gamma = 0.8$) | 12.08 | 0.38 | 0.49 | 51.5 | 96.8 % | 69.7 % |
| DINO w/o L1 ($\gamma = 0.8$) | 11.34 | 0.36 | 0.41 | 69.9 | 96.8 % | 69.4 % |
| DINO w/o L1 no balance fixed margin ($m = 0.2$) | 10.25 | 0.31 | 0.31 | 89.5 | 93.3 % | 58.8 % |

## A.4   ADAPTIVE BALANCING

As mentioned in Section 3 DINO uses a controller to ensure that the energy of generated samples is always a fixed multiple of the energy of real samples. Although this approach is similar to that used by BEGAN (Berthelot et al., 2017) there is a key difference. BEGANs perform autoencoding and therefore assume that the discriminator's reconstruction error will be larger for real samples since they have more details which are harder to reconstruct. For this reason, the controller used by BEGAN tries to maintain a balance throughout training where $\mathcal{L}(x_{real}) > \mathcal{L}(x_{fake})$. In the DINO framework the discriminator performs domain translation therefore it is natural to assume that real samples should produce better reconstructions since they contain useful information regarding the semantics. For this reason we choose to maintain a balance where $\mathcal{L}(x_{fake}) > \mathcal{L}(x_{real})$. This is reflected in the controller update as well as the balance parameter of DINO which is the inverse of that used in BEGANs.

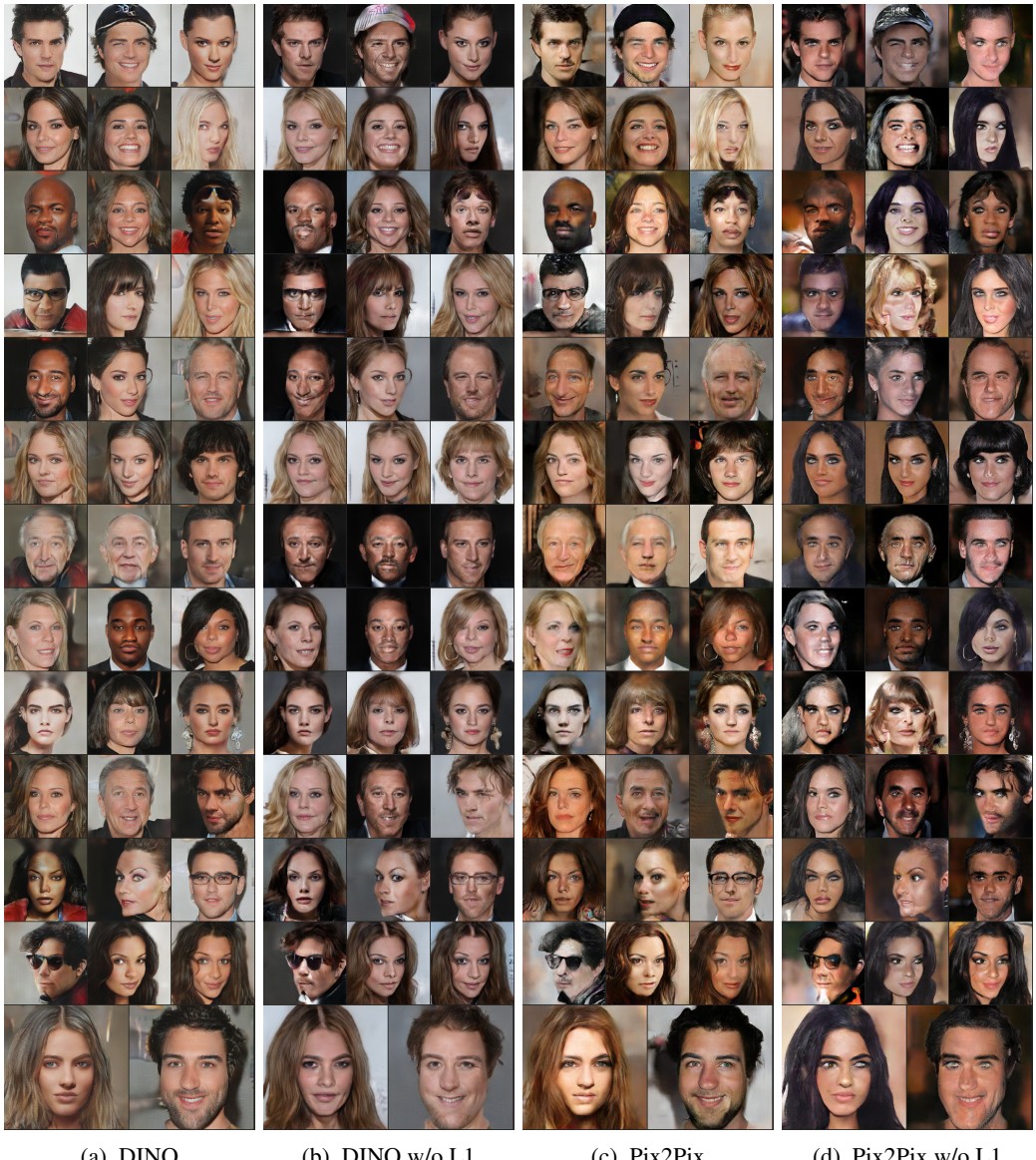

|  (a)  DINO  |  (b)  DINO w/o L1  |  (c)  Pix2Pix  |  (d)  Pix2Pix w/o L1 |

Figure 9: Examples of images generated with and without an L1 reconstruction loss. The L1 loss improves diversity both in the case of Pix2Pix and our proposed approach. Models trained without this loss will generate faces with similar attributes (e.g. hair color, skin color, age).

As we mentioned the core difference with the adaptive balancing in BEGAN is that DINO maintains a balance where $\mathcal{L}(x_{fake}) > \mathcal{L}(x_{real})$ whereas BEGAN maintains a balance for which $\mathcal{L}(x_{real}) > \mathcal{L}(x_{fake})$. This makes BEGAN unsuitable for use with the predictive conditioning proposed in this paper since it allows the generator to "hide" information about the condition in synthesized samples. The generator thus tricks the discriminator into producing a much better reconstruction of the condition for fake samples without the need for them to be realistic. Since the controller prevents the discriminator from pushing fake samples to a higher energy than real samples (i.e. the controller output is zero when fake samples have higher energy) this behaviour is not prohibited by BEGANs throughout training.

The method used in DINO however does not have this problem since it encourages the discriminator to assign higher energies to unrealistic samples thus penalizing them and preventing the generator from "cheating" in the same way as BEGAN. To show this effect we train a conditional BEGAN

and the DINO framework to perform translation from photo to sketch using the APDrawings dataset from (Yi et al., 2019). Figure 10 shows how the balancing used in DINO allows the network to penalize unrealistic images by encouraging the discriminator to assign to them energies larger than the real samples. We note that this problem occurs only in cases where the source domain is more informative than the target domain (i.e. photo → sketch). This does not occur in cases where the source domain in more generic than the target domain (i.e. segmentation map → photo)

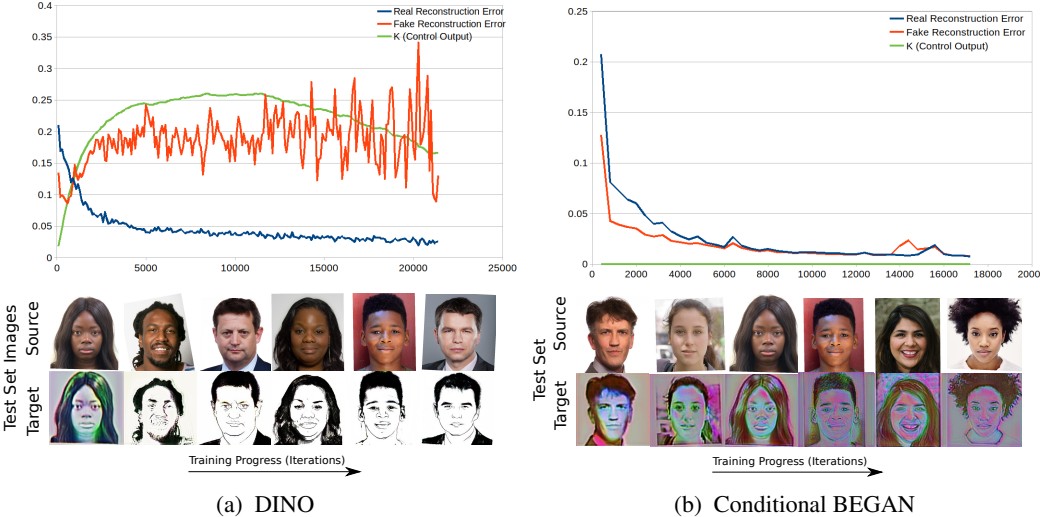

(a) DINO

(b) Conditional BEGAN

Figure 10: Example showing how the balancing used in BEGAN fails to produce realistic images. This happens because BEGAN does not incentivize the discriminator to assign larger energies to the fake samples. The figures below the plots shows how the performance on the test set progresses during training for each method.

## A.5 QUALITATIVE RESULTS

### A.5.1 IMAGE-TO-IMAGE TRANSLATION

**CelebAMask-HQ**

Examples of image-to-image translation from segmentation maps to photos for the CelebMask-HQ dataset are shown in Figure 11. We note that our approach is able to maintain semantics and produce realistic results even in cases with extreme head poses and facial expressions.

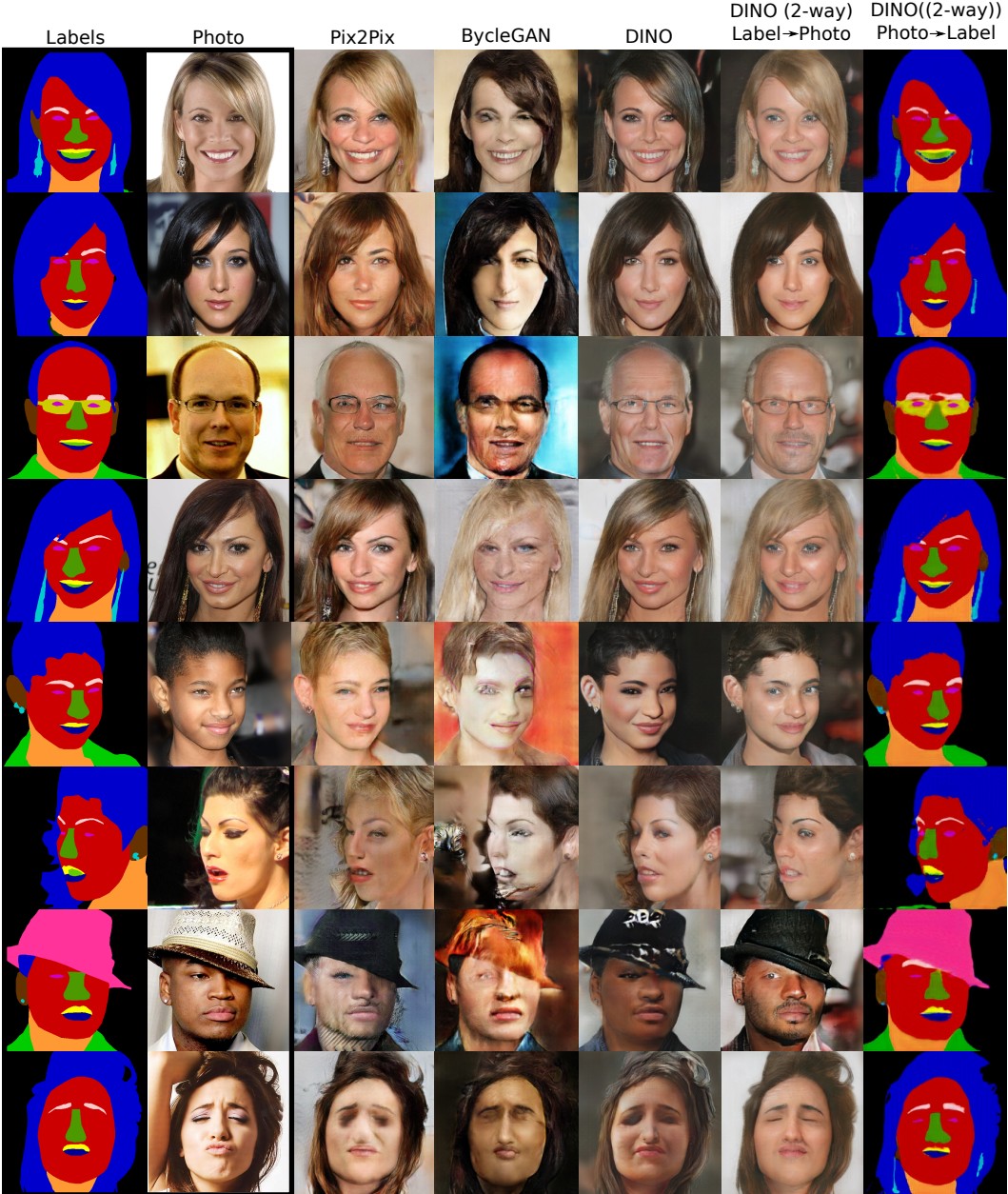

Figure 11: Image translation performed on the CelebAMask-HQ dataset. Translation is performed in the direction label → photo.

## Cityscapes

Examples of image-to-image translation from segmentation maps to photos for the Cityscapes dataset are shown in Figure 12.

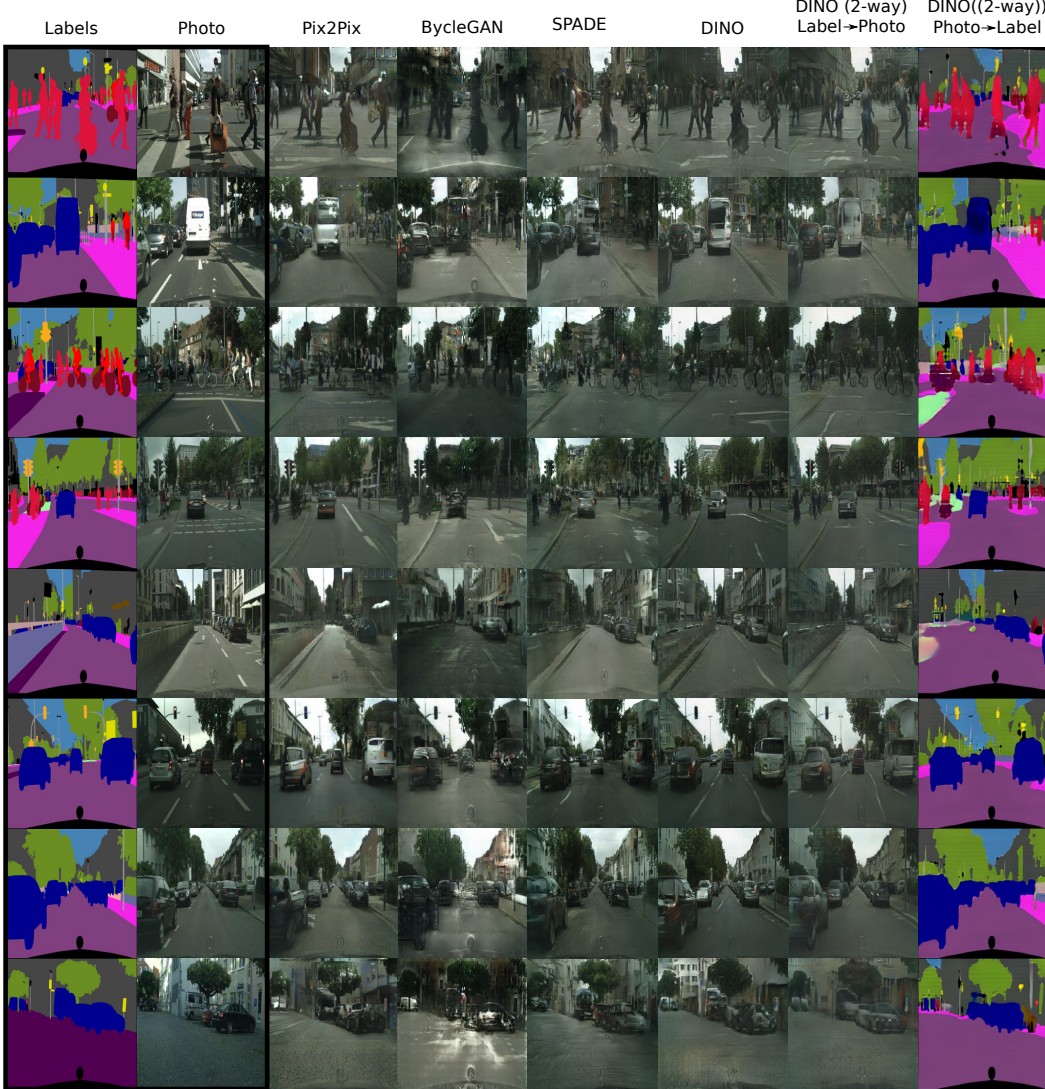

Figure 12: Image translation performed on the cityscapes dataset. Translation is performed in the direction label → photo.

## A.6 VIDEO-TO-SPEECH TRANSLATION

This section presents examples of waveforms produced by the methods compared in Table 3. In addition to the waveforms we also present their corresponding spectrograms. The waveforms and spectrograms are shown in Figure 13. It is evident from the shape of the waveform that our method more accurately captures voiced sections in the audio. Furthermore, the spectrogram of our method is closely resembles that of the ground truth although some high frequency components are not captured. The performance is similar to the Perceptual GAN proposed by Vougioukas et al. (2019) although our method relies on only an adversarial loss.

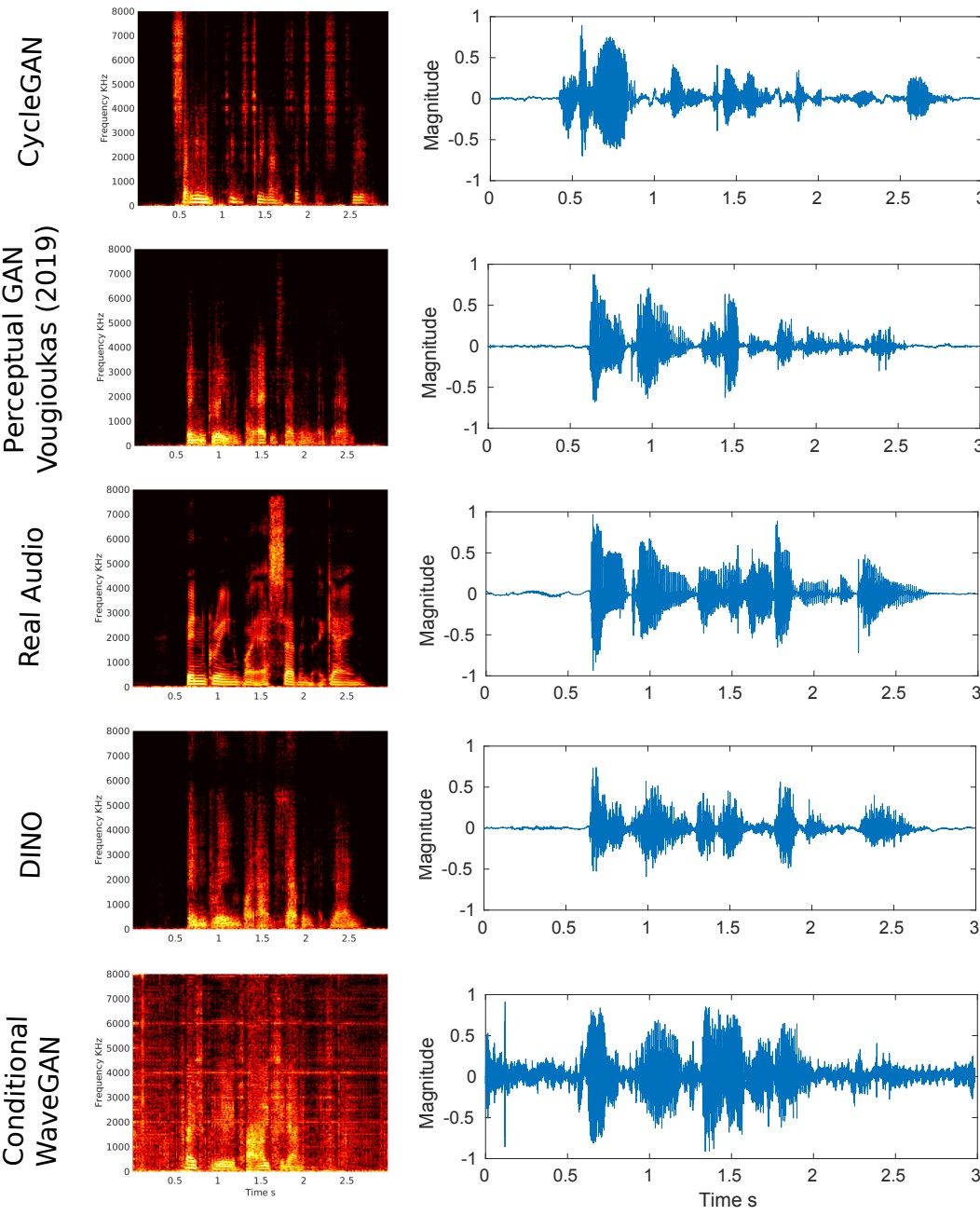

Figure 13: Examples of generated waveforms and their respective spectrograms. The real waveform and spectrogram is presented for comparison.

