# OpenReview forum: "DINO: A Conditional Energy-Based GAN for Domain Translation"
_ICLR.cc/2021/Conference — ICLR 2021 Poster_

### Official Review · AnonReviewer1 · 2020-10-27

**Rating:** 7
**Confidence:** 2

**Review:**

This paper presents a method for performing cross-domain or cross-modality translation models using a GAN-flavored framework where two models are trained to translate in both directions simultaneously.

[As a caveat: I am not well-versed in this area of the literature]

The paper is well-written for the most part and the experimental results are promising. My main concern are the ethical implications of some of the lip-reading experiments which go unaddressed.

**Pros**
- Clear presentation (mostly, see remarks for some exceptions)
- Good results as far as I can tell (although it is hard to interpret what all the various metrics mean, but it does seem like DINO is consistently better than the alternatives along most metrics)
- Experiments are not limited to image to image translation but also to "cross-modality translation" (image to text)

**Cons**
- If I understand correctly, the video->speech task is essentially a lip-reading task. State-of-the-art lip reading raises a number of privacy related concerns, and I think the potential impact of this research should at least be acknowledged in the paper.


**Remarks**
- The proposed method bears some conceptual similarity with recent work in unsupervised machine translation (see eg. Artetxe et al. https://arxiv.org/abs/1710.11041, Lample et al. https://arxiv.org/abs/1711.00043, also Lample et al. 2019 https://openreview.net/pdf?id=H1g2NhC5KQ which is particularly relevant as it tackles "style transfer" for text). These similarities are worth mentioning in the paper.
- I found the use of the term "discriminator" confusing, especially in the beginning in the paper. It makes sense in the usual GAN setup where the discriminator is an actual discriminative model, but it seems inappropriate in this case where the "discriminator" is a generative model.
- Eq. 6 (the main objective) is incredibly confusing. First, (and this relates to my last remark) the notation D_disc, D_gen, G_disc, G_gen unnecessarily confounding. Consider using "s -> t" and "t->s" for source to target and vice-versa instead of G and D. Also, perhaps color-coding Eq.6 would make it easier to parse (or maybe just separate the different terms a bit more).

---

> ### Author Response · Authors · 2020-11-14
> **Response to R1:**
>
> We thank R1 for his comments and for bringing our attention to the privacy concerns regarding the lip-reading application. We acknowledge this in an ethics statement which we have added to Section 4.2 par.3 of the paper. Please find our answers below:
>
> 1. We thank the reviewer for bringing these papers to our attention. They are indeed certain aspects that are similar in these approaches (i.e. back-translation, and bidirectional translation) hence we have mentioned and reference them in section 3.2.
>
> 2. We understand that it may be confusing to the reader to refer to the network as discriminator, especially in the bidirectional case where both networks essentially have both roles. We have changed the text by naming the translation networks Forward (source-to-target) and Reverse (target-to-source).
>
> 3. We updated the notation in the equation (now equation 7) to reflect the new naming of the networks and added underbraces to show which parts of the objective correspond to each network's role as a generator and which parts correspond to its role as a discriminator.

---

> > ### Comment · AnonReviewer1 · 2020-11-24
> > **Thank you for the response**
> >
> > I thank the author for the response.
> >
> > I appreciate that the addition of an ethics statement to address my main concern. The additional discussion with R3 regarding relevant baselines is also comforting.
> >
> > With the caveat that I am not very familiar with this general area of the literature, I maintain my recommendation that this paper should be accepted

---

### Official Review · AnonReviewer3 · 2020-10-28
**Seems reasonable, some clarity issues and not clear comparison is extensive (low confidence)**

**Rating:** 7
**Confidence:** 2

**Review:**

This paper proposes a conditional energy-based GAN technique for translation between data domains.

First, let me preface this review by noting that this paper is far outside of my area of expertise. I have tried to do my best in reviewing it, but I'd appreciate any clarification of mistaken points from the authors.

Overall, the idea itself seems reasonable: in conditional GAN-based models, instead of using a discriminator that explicitly tries to predict whether the generated output is true or fake, the discriminator tries to maximize the reconstruction score of true outputs, and minimize it for fake outputs.

However, there were many questions I had based on just reading the paper. I'm not sure whether this is due to my lack of background knowledge in this field, or because the writing itself is unclear (perhaps a bit of both):

1. First big question: from my reading it seems that this is a *supervised* model, in that it needs $(x,y)$ pairs to calculate the objective in Equation (2). Is this correct? It doesn't seem to be explicitly stated anywhere.

2. Given this, I was not sure if the baselines in Table 1 and 2 actually represent the state-of-the-art in this domain. Pix2pix seems to be from 2017, which seems to be quite old given the huge progress this field has made in the past 3 years. BiCycleGAN, according to my understanding, is an unsupervised method, which presumably will do much worse than supervised methods.

3. Are the datasets in 4.1 standard and used in the literature? If so, what are some recent papers that evaluate on these datasets? If not, why use these datasets instead of others? While I'm not very familiar with the field, I do know that image style transfer is a big thing, and surely there are other datasets that people have evaluated on previously.

4. The description of Equation (1) was a bit hard to follow, as the role of the discriminator was not made explicit. The "margin loss" was also not explained concretely.

5. I was not able to understand the description of $\gamma$ in Equation (3), please elaborate if possible. The gain value $\lambda$ was also not easy to follow.

6. It was mentioned that MirrorGAN is the most similar method. While there was an explanation that DINO is simpler than MirrorGAN, it would be nice to explain the implications of this. Does this just mean that DINO is a bit easier to implement? Or does it mean that it is fundamentally applicable to a wider variety of tasks? Also, why is there no empirical comparison with MirrorGAN?

---

> ### Author Response · Authors · 2020-11-14
> **Response to R3:**
>
> We thank R3 for his feedback and for taking the time to carefully read the paper. Here are the answers to the questions:
>
> 1: The reviewer is correct to say that this is a supervised method. We have amended the paper to explicitly mention this both in the abstract and when introducing the method.
>
> 2: Pix2Pix and BicycleGAN (this is stated in Section 3 of Zhu et.al. which mentions the need for paired data) are both supervised methods for image-to-image translation. We have chosen these methods because they are representative of supervised methods that use cGANs and VAE-GANs respectively. Although improved versions for Pix2Pix exist such as Pix2PixHD they rely on different architectures and training schemes (progressive growing, multiscale discrimination) to improve the resolution of images. Other state-of-the-art approaches like GauGAN (SPADE) (works only with segmentation maps), MaskGAN (works only on facial images) or APDrawing GAN (only performs sketch generation from photos) achieve impressive results but are tied to very specific types of translations. In this paper, we propose a generic domain translation method and a novel way of conditioning and wanted to compare it to standard conditioning methods for GANs. By keeping the network architecture simple and consistent for all models we can measure the effectiveness of predictive conditioning. We avoid comparisons with methods that are tailored to specific tasks since they often use additional information (pretrained models, accurate segmentation maps) to achieve their results. We have clarified this point in Section 4.1 of the paper.
>
> 3: Our method requires paired data and hence the selection of datasets is limited compared to unsupervised methods. We have chosen the Cityscapes and  CelebA-MaskHQ datasets because they are large, have high-resolution images and have previously been used for translation tasks. The cityscapes dataset has 5000 images with annotations and is used in many image translation papers including Pix2Pix, Pix2PixHD, and SPADE. The CelebA-MaskHQ contains 30,000 high-resolution face images selected from the CelebA with their corresponding annotations and is introduced and used in MaskGAN. We have chosen this because it is an annotated subset of CelebA, which is one of the most widely used datasets for faces. Finally, we use the APDrawings dataset in the Appendix (Fig.10) to show an example of why the balancing used in DINO is better suited for predictive conditioning compared to the balancing proposed in BEGAN. This is a small dataset of 70 images (when not counting augmentations by affine transforms) introduced and used in APDrawingGAN for sketch generation.
>
> 4: The margin loss refers to a loss which tries to maximize the energy of fake samples and minimize the energy for real samples. This is described in LeCun et.al. 2006, which mentions that a generalized margin loss is one that "uses some form of margin to create an energy gap between the correct answer and the incorrect answers". In the margin loss the fake samples are pushed to a high energy which is dictated by the margin parameter (highest allowable energy). This is a similar loss to that used in EBGAN and we have revised Eq.1 and added a new equation showing the exact formula for the loss (Eq. 2) to make this clear. The margin parameter in this loss (highest energy for fake samples) is a fixed number. In DINO we do not use a fixed margin with decay but use adaptive balancing instead (see next bullet-point)
>
> 5. As shown in EBGAN better results are achieved when the margin (as explained above) decays during training using some decay schedule. In order to avoid the need for a decay schedule, we use adaptive balancing to request that the highest energy of fake samples must always be a multiple of the energy of real samples E(Real)/γ =  E(Fake) (0<γ<1). The parameter γ is just dictating how large that multiple will be. For example, if gamma is 0.5 then we require that fake energies always be pushed so that they are 2 times the energies of real samples. To maintain this balance during training we use a proportional controller as is done in BEGAN. The controller affects how much emphasis the discriminator places on pushing fake samples to high energy. As fake samples get close to the high energy (E(Real) / γ ) the controller will slowly stop placing emphasis on pushing them to higher energy. The rate at which the controller changes the emphasis is the control gain λ.
>
> 6: MirrorGAN is indeed similar to our approach in that it enforces the semantics in generated image from text through redescription (image-to-text). However, unlike our approach, the network used for re-description is pretrained and used in addition to the discriminator of the GAN. In our method We have not compared to MirrorGAN because it is a GAN that has been designed for text-to-image synthesis and it is not straightforward to apply it for different translation tasks.

---

> > ### Comment · AnonReviewer3 · 2020-11-23
> > **Thank you for the clarification, but still concerns remain**
> >
> > Thank you for the clarifications here.
> >
> > This answered many of my questions, but in fact it somewhat increased my concerns regarding the empirical evaluation, as apparently there are (somewhat as I expected) many state-of-the-art approaches that have been excluded from the empirical comparison. While I understand the argument that more general approaches are preferable, if more specific approaches exist I feel they should be reported (maybe delineated as being more specific in the results tables). Right now it feels like the reporting is hiding the fact that the proposed models are far from the state of the art.
> >
> > Similarly, the fact that the method is supervised somewhat decreases its applicability obviously. Not that this is a big problem, but as the authors state the number of datasets where this supervised data is available (or could even feasibly ever *be* available) is quite limited.

---

> > > ### Author Response · Authors · 2020-11-23
> > > **Addressing remaining concerns**
> > >
> > > We thank the reviewer for the points raised in the discussion. We hope that the following sections will sufficiently address his concerns.
> > >
> > > 1) We avoid the comparison with certain models not because our approach is inferior in performance but because we believe that a comparison between methods should be done under similar conditions (e.g. architecture, resolution, required inputs). In fact when we compare our model's performance to the results reported for GauGAN and Pix2PixHD (100M parameters in the generator) in Park et.al 2019 we see that our method (37M parameters in generator) has higher segmentation accuracy (91.4% vs 81.9%) on the Cityscapes dataset and slightly worse FID score (51.5 vs 42.3) on the CelebAMask-HQ dataset. However, this is not a fair comparison because other methods usually make use of additional information (i.e. segmentation labels) or use losses that are tailored to a particular problem (whereas our approach is task-independent). Moreover, for some methods, a comparison is not possible since they are designed for different tasks. For example, MaskGAN deals with face editing, which takes as input a face, its segmentation map, and a user-modified segmentation map to produce an altered image. More importantly, all of these methods still use the same conditioning mechanism in the discriminator as proposed in cGAN. Since our goal is to evaluate the impact of the proposed predictive conditioning it is not useful to introduce more factors of variation in the comparison. The advantages of predictive conditioning are evident when we compare models with similar architectures (Table 1 and 2 of the paper).
> > >
> > >  We have also performed additional experiments to compare to the recently proposed NICE-GAN (CVPR 2020) using the official publicly available source code. The results are shown in the following table for the CelebAMask-HQ and Cityscapes dataset. Although this approach uses unpaired data and is expected to have a worse performance with regards to maintaining the semantics (i.e. Pixel. Acc. and mIoU) we note that it also has worse performance across all other metrics including perceptual metrics that should not be affected by the absence of supervision (i.e. FID and CPBD). This reveals that although NICE-GAN has seen the same training images from each domain (independently) as our method it still produces blurrier, less realistic results while using a far larger network and requiring longer training times (15 days vs 3 days on the same machine).
> > >
> > >  We would also like to point out that in the case of the speech reconstruction task we do compare to the state-of-the-art approach (Vougioukas et.al 2019) for video-to-waveform conversion and beat it despite its use of a pre-trained model for preserving content. This is shown in Table 3 of the paper.
> > >
> > >  We would like to stress that DINO is a generic domain translation method hence it is possible to apply it to a wide range of different tasks. We have chosen to showcase the performance for two very different translation tasks and we have shown that our model can perform well in both cases while remaining agnostic to the domains (i.e. does not use domain-specific knowledge). To the best of our knowledge, DINO is the first model to perform well for varied translation tasks whereas other generic translation methods such as CycleGAN and cGAN fail (as evident by the high WER in the speech-reconstruction experiment).
> > >
> > > |   Dataset|   PSNR| SSIM  | CPBD  |  FID |   Pixel Acc. |  mIoU |
> > > |---|---|---|---|---|---|---|
> > > |  CelebA |10.24   |  0.31 |  0.4 | 53.20 | 93.4%| 62.9%  |
> > > | Cityscapes | 13.89 |  0.30  | 0.59    | N/A   | 65.7% | 17.9%  |
> > > | |Table: |NICE-GAN | evaluation | | |
> > >
> > > 2. Although we agree that the need for paired examples does indeed restrict supervised methods there are still many datasets that have paired data. Indeed many successful approaches in image-to-image translation require supervision (Pix2Pix, Pix2PixHD, GauGAN). This is also especially true for cross-modal translation systems which are still mostly supervised (Qiao et al., 2019, Vougioukas et al. 2019). Finally, for some problems such as speech-driven animation or video-driven speech reconstruction the data from two domains is abundant and comes naturally paired (i.e. the video is recorded with its corresponding audio).

---

> > > > ### Author Response · Authors · 2020-11-24
> > > > **Update: Comparison with Task-Specific SoTA Pretrained Gaugan (SPADE)**
> > > >
> > > > To further alleviate the reviewer's concerns we have taken the official source code and pre-trained model for SPADE (CVPR 2019) for the Cityscapes dataset and calculated all our metrics for the generated images. We have added these in the paper so that there is now a direct comparison with a state-of-the-art task-specific model.  The results are shown in the following Table which we have also added to the main paper (Tabe 2).
> > > >
> > > > | Model  | PSNR   | SSIM   | CPBD   |Pixel. Acc   | mIoU |
> > > > |---|---|---|---|---|---|
> > > > |  SPADE | 15.97  |  0.38 | **0.72**  |**92.8 %** | **38.3 %** |
> > > > |  DINO | 16.44 |  0.41 | 0.71  | 91.4  %|35.9 %|
> > > > |  DINO (2-way)| **16.78** |  **0.42** |**0.72**  | 91.3  %|35.2 %|
> > > > ||Table:|Comparison with SPADE on Cityscapes||||
> > > >
> > > > As is evident from the results, our model is comparable to the SPADE model. As noted in the previous discussion the SPADE model is specifically designed for translation from segmentation maps whereas our model is generic (not limited to one type of translation) and can perform the translation in both directions.
> > > >
> > > > Also, we would like to point out that for the more challenging video-to-waveform translation we are outperforming the task-specific state-of-the-art model (Perceptual GAN) as shown in Table 3 the main paper.

---

> > > > > ### Comment · AnonReviewer3 · 2020-11-24
> > > > > **Thank you for the clarification and updated results**
> > > > >
> > > > > Thank you, this addresses my major concern.
> > > > >
> > > > > I'd like to reiterate that I'm still not very confident in my assessment here, but based on the revision I think that the paper probably warrants acceptance and will raise my score accordingly.

---

### Official Review · AnonReviewer2 · 2020-10-29
**Energy based GAN with symmetric generators**

**Rating:** 7
**Confidence:** 3

**Review:**

The paper proposes an adversarial framework DINO to train translation models from source to target and target to source. The basic idea is to replace generator and discriminator in the energy based GAN with two source-to-target generation models. The discriminator(reverse generator) and the generator competes in a minimax game to reconstruct the data. The framework is further extended with duplicate output heads for both discriminator and generator to enhance the training robustness.
The authors evaluated their framework on two tasks: image to image translation and silent-video to speech reconstruction. The DINO method impressive improvement in both tasks.

Strong points:
1. The proposed DINO framework is well motivated. The objectives in DINO are reasonable and novel.
2. Experiments on both image to image translation and video to speech reconstruction verify the DINO method achieves significant improvement comparing with other translation methods.

Weak points:
1. Important details are omitted in image-to-image translation and video to speech reconstruction. It is unclear about the backbone network as well as the parameter setup. Therefore, it is impossible to reproduce the method.
2. DINO and DINO(bidirectional) are not consistent winners. It is not explained or analyzed why DINO sometimes wins while DINO(bidirectional) wins otherwise. There is no recommendation for practical use either.
3. The adaptive balancing seems reasonable. But it is not studied in the experiment whether it improves the training.

---

> ### Author Response · Authors · 2020-11-14
> **Response to R2:**
>
> We thank R2 for his feedback and comments. Please find our answers below:
>
> 1: We realize that some details such as the optimizer and learning rate used for the experiments are missing from the paper. We thank the reviewer for pointing this out. In the image-to-image experiments, we used an Adam optimizer with a learning rate of 0.0002 and a batch size of 8. In the audio-visual experiments, we used an Adam optimizer with a learning rate of 0.0001 for the parameters of the Video-to-Audio network and a learning rate of 0.001 for the parameters Audio-to-Video network. We used a batch size of 8 for the Video-to-Audio experiments. We have added this information in the paper in Sections 4.1 and 4.2. The details of the architecture for each network are shown in the Appendix in Section A.1. We will also make our source code and pretrained models publicly available so that others can reproduce the results.
>
> 2: The reviewer is correct to say that there is not a clear winner in terms of performance between the bidirectional and unidirectional version of DINO. In the unidirectional version, the network acting as the discriminator will not benefit from the adversarial loss and will therefore produce less realistic results. To obtain realistic samples in both directions using the unidirectional DINO method training would have to be performed separately for each direction and the discriminator network would be discarded each time. This problem can be solved by using the bidirectional version where both networks benefit from an adversarial loss and both produce realistic samples (and no network is discarded after training). Therefore, the bidirectional DINO method can train for bidirectional translation in half the time of the unidirectional version while achieving comparable results. The motivation for bidirectional translation is to reduce the number of parameters required for bidirectional translation without sacrificing the quality of the results. This is reflected in section 3.1 where we introduce the bidirectional model and in the conclusion.
>
> 3: DINO can work without the adaptive balancing by using a fixed margin for the margin loss like in EBGAN. However, we have found that using adaptive balancing improves the stability and quality of the results. The performance with and without adaptive balancing is shown in our ablation study in the Appendix Section A.3 and Table 4 (rows 2 and 3). The results show that adaptive balancing results in a significant improvement across all performance metrics.

---

### Author Response · Authors · 2020-11-24
**Summary of Changes**

We thank the reviewers for their feedback. We have compiled a comprehensive list of changes that we have made to the rebuttal revision (compared to the original submission) following the discussions:

1. We explicitly state that the DINO method is supervised (i.e. requires paired data) in the abstract to avoid confusion.
2. We have changed the names of our networks from generator and discriminator to forward and reverse translation networks as per R1's request and updated the notation in equations (Eq.3-7) to match.
3. We have included the related works suggested by R1 in section 3.2 (paragraph 3).
4. We have added a discussion about the SPADE method in Section 2.1 (paragraph 2) and also added SPADE to the list of methods that we compare against in the Cityscapes dataset (Table 2, Figure 12 in the appendix).
5. We have added descriptions for the margin loss and added an additional equation (Eq.2) for clarification.
6. We have added more details regarding hyperparameters used for training our model in Sections 4.1 (paragraph 4) and 4.2 (paragraph 3).
7. We have included the source images in Fig. 10 of the Appendix so that the readers can see the type of data used to drive the translation.

---

### Decision · Program_Chairs · 2021-01-07
**Final Decision**

**Decision:**

Accept (Poster)

**Comment:**

This paper presents a novel method for general-purpose supervised domain transfer that trains both generator and discriminator to compete in a minimax game in order to reconstruct data. This setup is meant to address a common issue in conditional GAN setups: they often ignore conditioning information. Results are positive and span two very different tasks: image-to-image translation and silent-video-to-speech reconstruction. Overall reviewers were quite positive about this paper: they found the method to be novel and well-motivated, and after rebuttal, found experimental results to be sufficiently convincing. Several concerns were brought up: (a) lack of emphasis that the approach is in fact supervised, (b) need for comparisons with stronger or task-specific baselines, (c) lack of description of experimental details for reproducibility, and (d) lack of discussion of ethical implications. All of these concerns were satisfactorily addressed by authors in rebuttal and reviewers unanimously vote for acceptance. I agree, and recommend this paper be accepted.